# Relationship between Diet Quality and Socioeconomic and Health-Related Factors in Adolescents by Gender

**DOI:** 10.3390/nu16010139

**Published:** 2023-12-30

**Authors:** Ester Colillas-Malet, Marina Bosque-Prous, Laura Esquius, Helena González-Casals, Aina Lafon-Guasch, Paula Fortes-Muñoz, Albert Espelt, Alicia Aguilar-Martínez

**Affiliations:** 1Epi4health Research Group, Faculty of Health Sciences of Manresa, Universitat de Vic—Universitat Central de Catalunya (UVic-UCC), Av. Universitària 4-6, 08242 Manresa, Spain; ecolillas@umanresa.cat (E.C.-M.); hgonzalez@umanresa.cat (H.G.-C.); alafon@umanresa.cat (A.L.-G.); albert.espelt@uab.cat (A.E.); 2Epi4health Research Group, Faculty of Health Sciences, Universitat Oberta de Catalunya, Rambla del Poblenou, 156, 08018 Barcelona, Spain; mbosquep@uoc.edu (M.B.-P.); pfortesm@uoc.edu (P.F.-M.); 3Departament de Psicobiologia i Metodologia en Ciències de la Salut, Universitat Autònoma de Barcelona (UAB), C/de Ca n’Altayó s/n, 08193 Bellaterra, Spain; 4Centre d’Estudis Epidemiològics sobre les ITS i Sida de Catalunya (CEEISCAT), 08916 Badalona, Spain; 5Fundació Institut d’Investigació en Ciències de la Salut Germans Trias i Pujol (IGTP), 08916 Badalona, Spain; 6Centro de Investigación Biomédica en Red de Epidemiología y Salud Pública (CIBERESP), C/Monforte de Lemos 3 Pabellón 11, 28029 Madrid, Spain; 7Foodlab Research Group, Faculty of Health Sciences, Universitat Oberta de Catalunya, Rambla del Poblenou, 156, 08018 Barcelona, Spain; aaguilarmart@uoc.edu

**Keywords:** adolescents, diet quality, socioeconomic factors, health behavior, social inequalities

## Abstract

Adolescence is a key period for consolidating heathy lifestyles and proper eating habits that can last into adulthood. To analyze the diet quality of Spanish adolescents and its association with socioeconomic factors and health behaviors by gender, a cross-sectional study was conducted using data from the DESKcohort project, consisting of a biannual panel survey on health, health behaviors, and associated determinants, collected in secondary education centers. The study population consisted of 7319 students aged 12 to 18 years. Data were collected from October 2019 to March 2020. The dependent variable was diet quality score according to the Spanish adaptation of the Healthy Eating Index (S-HEI). The independent variables included were socioeconomic factors and health behaviors. We conducted linear regression separately by gender. Diet quality score was significantly higher for girls than for boys (68 and 65, respectively, *p* < 0.001). For both genders, poorer diet quality was associated with a low level of physical activity [−0.9 (95% CI = −1.6:−0.2) in boys, −1.2 (95% CI = −1.9:−0.4) in girls], alcohol use [−2.5 (95% CI = −3.7:−1.3) in boys, −1.0 (95% CI = −1.9:0.0) in girls], poor self-perceived health [−1.1 (95% CI = −2.4:0.2) in boys, −3.5 (95% CI = −4.6:−2.4) in girls], and having attended Intermediate Level Training Cycles [−2.9 (95% CI = −4.3:−1.5) in boys, −1.9 (95% CI = −3.5:−0.3) in girls]. In girls, poorer diet quality also was associated with low mood [−1.1 (95% CI = −1.9:−0.3)]. The variance was 9% in boys and 12% in girls. Our results highlight the need to consider socioeconomic and health-related factors, as well as gender, when conducting interventions to promote healthy eating among adolescents.

## 1. Introduction

Adequate nutrient intake and adherence to a healthy diet are important factors to stay healthy and prevent diseases [1]. Adolescence is a period in which children gain increasing control over their food choices and dietary habits. Therefore, this is a very suitable period to establish healthy lifestyles that can persist into adulthood, and an ideal stage for the implementation of health promotion programs that can impact individuals’ present and future [2,3].

In Spain, more than 69% of the adolescent population has a suboptimal adherence to the healthy eating pattern characteristic of their context, the Mediterranean Diet [4]. Similar percentages have been found in other countries in the Mediterranean region [3,5]. Studies have also shown a lower than recommended consumption of fruit, vegetables, and dairy products and a higher than recommended consumption of meat and meat products, fats, and sweets [6].

Diet quality indices allow for the categorization of dietary patterns as more or less healthy and helps determine risks for chronic non-communicable diseases. The Healthy Eating Index (HEI) is one of the most widely referenced and validated indices on diet quality [7]. Higher scores have been associated with a lower risk of all-cause mortality, cardiovascular diseases, some types of cancer, type 2 diabetes, and neurodegenerative diseases [8]. HEI has been successfully used to monitor diet in adolescents [9], and is also used to evaluate nutritional interventions and educational programs [10]. Adaptations on the HEI exist for dietary guidelines in different countries. In Spain, the Spanish Healthy Eating Index (S-HEI) was developed to measure adherence to healthy eating according of the Spanish Society of Community Nutrition [11].

The adaptation of the dietary quality indices to different countries means that the scores obtained between them are not comparable, however the periodical estimation of the quality of adolescents’ diet, its evolution, and the comparison of results between regions and over time can be of great interest to plan future food and nutrition national policies [12]. 

Socioeconomic factors and health behaviors are major determinants of dietary behaviors [13]. Previous studies [3,10] showed, for example, how individuals with more advantaged socioeconomic positions, with a higher level of education or those that are more physically active have healthier diets. Mood is also related to the consumption of certain foods, especially among women [14]. Women and elderly people also have a better diet quality [15,16] and gender differences in dietary behaviors can already be observed during adolescence [17,18]. In addition, interventions that require behavioral changes may benefit from considering possible gender differences since evidence suggests that gender is important in decision making, participation, communication, and preferences for acceptance of interventions [19]. 

Investigating the factors associated with diet quality may be of vital importance in adapting future interventions [20]. 

Thus, the objective of this study was to analyze diet quality in a sample of 12- to 18-year-old adolescents from Central Catalonia (Spain), and ascertain its association with demographic and socioeconomic factors, and with health behaviors by gender. 

## 2. Materials and Methods

We used a cross-sectional design to study data from the first wave of the DESK-cohort project [21], a cohort study consisting of a biannual panel survey on health, health behaviors, and associated determinants. The survey, which lasted a maximum of 45 min, consisted of a maximum of 69 questions, some of which were filtered; this survey was administered to 12- to 18-year-old adolescents schooled in Central Catalonia. Our data were collected in the period from October 2019 to March 2020. 

The study consisted of a convenience sample of students aged 12 to 18 years old attending school during the academic year 2019–2020 in Central Catalonia. Central Catalonia is an inland region with a combination of small urban and rural areas (<150 inhabitants/km^2^ or municipalities of less than 5000 inhabitants) in northeastern Spain [22,23]. A letter was sent to all educational centers (compulsory secondary education and post-compulsory secondary education centers) of the region (n = 91) inviting them to participate in the project, and a total of 65 (71.4% of the total) accepted. 

Participation was voluntary and participants were those present in class on the assigned data collection day for each secondary school. These participants did not receive any financial compensation. Each participant responded to the online self-administered questionnaire in a classroom using a tablet connected to the Research Electronic Data Capture (Redcap) system, following the Organic Law on the Protection of Personal Data regulation [24]. The initial database consisted of 8491 records. During debugging, 1172 were removed for being too markedly incomplete. Therefore, 86.2% of the participants completed the entire questionnaire.

This study was conducted according to the guidelines of the Declaration of Helsinki and all procedures were approved by the Ethics and Research Committee (blinded peer review). Written consent for participants aged 14 and under was obtained from parents or legal guardian and, for participants over 14, was obtained from the participants themselves.

Study instructors were trained on the questions and the variables of the study and were present in each classroom to answer participant questions. 

Data collection was conducted in the educational centers. Information on food consumption was collected via the food frequency questionnaire (FFQ) used was obtained from the survey on Risk Factors in Secondary School Students designed and extensively used for a similar population by the Public Health Agency of Barcelona, in Catalonia [25]. This questionnaire asks about the frequency of habitual consumption of 20 food groups. The following 7 response options were available: never; one to three times per month; once per week; two to three times per week; four to six times per week; once per day; and more than once per day. Subsequently, as a dependent variable, we assessed diet quality using the S-HEI [11], an adapted version of the original Healthy Eating Index [26]. The S-HEI includes information on the consumption of 10 items: vegetables, cereals, legumes, fruit, meat, milk and dairy, sweets, processed meats, soft drinks, and dietary variety [11]. For this study, the 20 food groups obtained from the FFQ were recoded to obtain 10 food items used in S-HEI. The following 5 response options were available: never or hardly never; once per week; once or twice per week; more than 3 times per week, but not daily; and daily. These options have been defined according to the frequency of food intake indicated within the guidelines of the Spanish Society of Community Nutrition (see Appendix A) [27] and obtained after recording the response options of the FFQ. Each food group includes various foods to which the participants assigned a frequency that was, in turn, associated with a value. Then, the values for each food were added up and given the total score for the group (ranging from 0 to 10). Finally, the total S-HEI score is the sum of the scores obtained for each of the food groups, plus 0 to 10 points based on compliance with the daily and weekly recommendations for the different food groups (dietary variety). A higher S-HEI score denotes greater adherence to the guidelines of the Spanish Society of Community Nutrition and, therefore, a higher diet quality. For the descriptive analysis, we analyzed diet quality in two ways: continuous and categorical. To obtain the latter, we divided the continuous S-HEI into three categories: unhealthy diet (<50 points); diet that needs changes (50–80 points); and healthy diet (>80 points) [11]. As the distribution in the categories was uneven, we performed the rest of the analyses with the continuous variable. 

The FFQ data were also used to describe whether the participants met the recommendations for frequency of consumption of different foods according to the recommendations of the Spanish Society of Community Nutrition [27]. The variables related to the consumption of different types of food were the following: (1) cereals; (2) vegetables; (3) fruit; (4) milk and dairy products; (5) nuts; (6) legumes; (7) meat; (8) fish; (9) eggs; (10) processed meats; (11) soft drinks; (12) energy drinks; (13) sweets; (14) pastries; (15) snacks; and (16) fast food. The consumption frequency of each type of food was classified as follows: lower than recommended, recommended, or higher than recommended, as indicated within the guidelines of the Spanish Society of Community Nutrition [27].

As independent variables, we considered the following demographic, socioeconomic, and health-related factors. The demographic variables were: (1) gender (boy/girl); (2) age (years of age determined from the date of birth); (3) course (second and fourth courses of compulsory secondary education (CSE), second course of post-compulsory secondary education (PCSE), and Intermediate Level Training Cycles (ILTC)). The equivalence of these levels according to the UNESCO International Standard Classification of Education (ISCED) corresponds to ISCED 2 for CSE and ISCED 3 for PCS and ILTC [28]; and (4) size of municipality (≤5000 inhabitants, 5001–20,000 inhabitants, or >20,000 inhabitants). The socioeconomic variables were (5) perceived socioeconomic position (SEP) (disadvantaged, medium, or advantaged, determined by using the MacArthur Scale of Subjective Social Status, on a scale from 0 to 100, where higher values indicated more advantaged SEPs. The continuous variable was categorized into tertiles) [29]; and (6) parents’ highest level of education (primary, secondary, or university education). The health-related factors studied were (7) self-reported weight and height used to calculate Body Mass Index (BMI), which was recoded into underweight, healthy weight, overweight, or obesity, as defined using age- and sex-specific BMI cut-offs according to the WHO growth reference for school-aged children and adolescents [30]. Other health-related factors studied were (8) physical activity (in compliance with the WHO recommendation of ≥60 min per day, or under the WHO recommendation of 60 min per day [31], estimated from the average daily minutes of moderate or vigorous physical activity reported by the adolescents); (9) self-perceived health (excellent/very good, good, or poor/very poor); (10) mood state (assessed through six questions [25] with answers from 1 (Never) to 5 (Always), then grouping the results in the two categories “never”/“almost never”/“sometimes” (value 0) and “often”/“always” (value 1), and finally adding up the scores for each item, with a final score of 3 or more identified as low mood [32,33]); (11) sleep quality (very good/good or poor/very poor); (12) alcohol use (hazardous drinking or non-hazardous drinking estimated using the Alcohol Use Disorders Identification Test—AUDIT-C test [34], with scores above 3 considered hazardous); (13) tobacco use (daily use or other); (14) cannabis use (risky consumption or non-risky consumption, estimated using the Cannabis Abuse Screening Test—CAST [35], with scores above 7 considered risky); (15) mobile phone use (occasional or frequent problematic use or non-problematic use, estimated using the questionnaire for mobile phone-related experiences—CERM test [36], with scores above 15 considered problematic); (16) self-reported academic performance (good grades, average grades, or poor grades); and (17) having experienced bullying (yes or no). 

### Data Analysis

Descriptive and inferential statistics were used to summarize and compare participant characteristics by gender. Frequencies and percentages were reported for categorical variables with chi square tests used to determine gender differences. Means and standard deviations were reported for continuous variables with 2-sample *t* tests used to determine gender differences. For categorical variables with missing data, a separate category was created to include these participants in the analysis. Affected variables were size of municipality, parents’ highest level of education, Body Mass Index, physical activity, mobile use, self-reported academic performance, and having experienced bullying. Multivariable linear regression models were built to determine significant explanatory variables for S-HEI using backwards elimination. All explanatory variables were initially included in the models and iteratively removed until only variables significant at the 0.05 level remained. Regression models were built separately for boys and girls based on expert recommendation [37]. Regression coefficients and corresponding 95% confidence intervals were computed for the final model. A sensitivity analysis was carried out using quantile regression models, which model the median instead of the mean. All statistical analyses were conducted with STATA 16.

## 3. Results

The study population consisted of 7319 students aged 12 to 18 years. Table 1 shows their socioeconomic characteristics and health behaviors by gender. A total of 52.1% of the study population were girls. We observed no difference in age between boys and girls (mean = 15.3 years and SD = 0.03 for both). In relation to health behavioral factors, around 21.7% of boys and 14.9% of girls were overweight or obese; around 37.4% of boys and 57.7% of girls did not comply with the WHO recommendations; around 10.5% of boys and 15.7% of girls engaged in hazardous alcohol drinking; around 6.7% of boys and 8.3% of girls smoked daily; and around 21.6% of boys and 26.5% of girls engaged in problematic use of mobile phones. In terms of self-perceived health, around 65.3% of boys and 50.1% of girls reported excellent/very good health, and around 12.3% of boys and 25.8% of girls had low mood. The mean diet quality score (S-HEI) was significantly higher for girls than for boys (around 67.6 vs. 65.1, *p* < 0.001).

Figure 1 shows the distribution of frequencies of consumption (lower than recommended; recommended; higher than recommended) of each food group according to the dietary guidelines of the Spanish Society of Community Nutrition by boys and girls. The distribution of consumption frequencies is significantly different between boys and girls (*p* < 0.05) in all food groups, except for fish consumption (*p* = 0.245). The proportion of boys consuming foods more frequently than recommended was higher than that of girls for most of the foods considered. Foods that boys and girls consumed more frequently than recommended were processed meats, pastries, snacks, and soft drinks. Foods that boys and girls consumed far less frequently than recommended were vegetables, cereals, fruit, and milk and dairy products (detailed percentages can be found on Appendix A).

In Table 2, we show the mean S-HEI scores for the categories of each independent variable by gender. We found significantly lower S-HEI mean scores (from worse to better S-HEI) in attending ILTC, having a disadvantaged SEP, having parents with primary education, not meeting WHO recommendations for physical activity, hazardous alcohol drinking, daily tobacco use, risky use of cannabis, occasional or frequent problems with mobile phone use, reporting poor/very poor self-perceived health, low mood, reporting poor/very poor sleep quality, having poor grades, and having experienced bullying.

Table 3 contains the results from the multivariable linear regression analysis for S-HEI. Among boys, the factors associated with S-HEI score were the following (from higher to lower βa): poorer grades in school (βa = −3.9); daily tobacco use (βa = −3.1); attending ILTC (βa = −2.9); hazardous alcohol drinking (βa = −2.5); occasional or frequent problems with mobile phone use (βa = −2.3); having parents with a primary education (βa = −2.1); reporting good self-perceived health (βa = −1.1); and not meeting WHO recommendations for physical activity (βa = −0.9). Among girls, factors associated with S-HEI score were the following (from higher to lower βa): poorer grades in school (βa = −5.2); reporting worse self-perceived health (βa = −3.5); having parents with a primary education (βa = −3.3); daily tobacco use (βa = −2.6); occasional or frequent problems with mobile phone use (βa = −2.0); attending ILTC (βa = −1.9); not meeting WHO recommendations for physical activity (βa = −1.2); and hazardous alcohol drinking (βa = −1.0). In addition, in girls, S-HEI scores were inversely associated with low mood (βa = −1.1). The variance was 8.9% in boys and 12.2% in girls. A sensitivity analysis was conducted using quantile regression models, and we obtained similar results (see Appendix A).

## 4. Discussion

Our findings show that 92% of the adolescents in the study need to improve the quality of their diet (S-HEI score ≤ 80 points). We found that diet quality was associated with the following factors: gender, level of physical activity, substance use (alcohol and tobacco), self-perceived health, parents’ level of education, academic course, academic performance, and problematic use of mobile phones. However, BMI was not related to diet quality. 

Studies on diet quality in adolescents from other countries also show far from optimal mean HEI scores [10,12].

There is a need to improve adolescents’ diet, in accordance with studies showing a gradual loss of healthy eating patterns in European countries [4]. We found that most adolescents eat fruit, vegetables, cereals, and milk and dairy products less frequently than recommended; moreover, most of them eat processed meats, pastries, soft drinks, and snacks more frequently than recommended. Adolescents consumed less than five servings of fruits and vegetables per day, with boys consuming less than girls. These findings were similar to previous research in Spain [38] and other countries [39]. Processed meats, pastries, and snacks are high in saturated fats, salt, and added sugars; therefore, they can contribute to the energy intake of adolescents, but have low nutritional value. Also, other studies in adolescents found that similar food groups are top contributors to the discretionary energy intake [40].

To improve diet quality, it will be necessary to increase the consumption of vegetables, fruit, cereals (preferably whole), and to decrease the consumption of processed meats, pastries, soft drinks, and snacks. The diversification of protein sources in favor of the consumption of legumes, fish, or eggs could contribute to a decrease in the consumption of meat and processed meat, with greater health and sustainability benefits, and would also be more aligned with the Mediterranean dietary pattern [41,42,43].

Consistent with previous research [10,11], girls showed better diet quality than boys: they are more strict about it [17] and tend to adopt more regular dietary behaviors. The latter could be linked to nutritional knowledge, awareness of their bodies, and different familial influences. For instance, modeling seems to affect girls more than boys [44]. As for adolescents’ mood state, some studies suggest that difficulties in regulating emotions is an underlying or common factor for different maladaptive behaviors [45]. Likewise, the relationship between low mood and lower S-HEI scores that we found in girls is consistent with studies showing a relationship between gender and emotional motivations in food choice to cope with stress, or as a comfort when depressed [14]. Therefore, a gender perspective in the design of interventions can contribute to improving their acceptability and efficacy [19].

The positive association between parental education and S-HEI score found in the current study coincides with others reporting that parenteral education plays a role in the adoption of healthier eating behaviors [46]. Specifically, parental education was associated with an increased consumption of fruits and vegetables [47]. Thus, adolescents with more highly educated parents may have a home environment more conducive to healthy eating. Research has shown that education is positively associated with diet quality in adults [48]. Hence, the eating habits modeled by more highly educated parents may positively affect the eating habits of their children, particularly regarding the consumption of fruits and vegetables [47,49]. Further, the higher income often associated with higher education may allow parents to provide healthier food options for their children as well as to live in neighborhoods more supportive of healthier eating [50]. 

Our results show an association between S-HEI score and poor academic performance: adolescents that had poorer grades in school had a lower S-HEI score. Therefore, providing tools to improve the quality of adolescents’ diet could not only produce nutritional benefits, but also cognitive benefits. Previous studies have found that dietary patterns high in vegetables and fruits have been directly associated with cognition, and dietary patterns high in fast food, processed meats, and soft-drinks have been inversely associated with cognition [51].

Although no association was found between diet quality and age, students in ILTC had the lowest S-HEI scores. These results are in line with those obtained from students in Central Europe where pupils of ILTC showed a higher prevalence of obesity, as well as a less healthy diet, in comparison to CSE students. ILTC in Spain is a post-compulsory education option for those students whose intention is not to go to university but to train in professions like trade, electricity, mechanics, or hospitality. Although further analysis of the demographic profile is needed [52,53], the evidence [54] indicates that ILTC is mainly attended by students from less favored socioeconomic positions and with a lower previous academic performance and these factors could overlap and influence the quality of their diets [55]. There is also evidence of a higher co-occurrence of other health risk factors, such as smoking, risky alcohol consumption, and less physical activity among ILCT students, makes them a particularly vulnerable group for which to develop holistic health interventions [56]. In this sense, previous studies suggest that the most popular health promotion programs for these students include hands-on activities such as cooking classes, increasing the availability or affordability of healthy food, and expanding the inclusion of peer-led activities, as individuals may model their behaviors on activities that are perceived to be socially acceptable [54,57].

Similarly, the association of lower S-HEI with substance use (alcohol and tobacco) and less physical activity highlights the need to plan health promotion programs as an integral part of the education program at ILTC and CSE centers [58]. 

Consistent with the existing literature [59,60], we found that S-HEI was positively associated with physical activity and self-perceived health. In line with this, the WHO identifies interventions on diet and physical activity as priority areas for promoting healthy lifestyles in educational environments [51].

The association between poor S-HEI scores and problematic use of mobile phones is consistent with previous research on mobile phone use and non-mobile media (TV or computer). Indeed, screentime has been directly associated with poorer diet quality, a more frequent consumption of soft drinks, more high-fat/high-sugar foods, and a lower consumption of vegetables and fruit [61]. 

In this study, relationships found among diet quality, socioeconomic factors, and health behaviors in both girls and boys highlight the need to continue efforts to improve diet in Spanish adolescents. This may be particularly relevant for adolescents in CSE, PCSE, and ILTC. Schools can promote an environment that encourages healthy eating, but it is also important to include nutritional education in the syllabus, along with the teaching of skills and attitudes. Moreover, such strategies must pursue sustainable long-term impacts to influence the future health of the population [20]. 

CSE, PCSE, and ILTC interventions should also provide families with information to facilitate habit changes and raise awareness regarding the importance of maintaining a healthy diet. Conducting family sessions or workshops to empower parents with practical knowledge of nutrition and healthy cooking (information on dietary recommendations or how to include healthy ingredients in teens’ favorite recipes, for example) can be key facilitators in promoting healthy eating at home along with involving adolescents in meal preparation and encouraging family discussion about food at mealtimes [62].

Moreover, different complex factors, such as peers or advertising, may negatively affect the eating habits of adolescents. Schools need to create an integrated healthy nutrition environment [17]. 

This study has several limitations. First, the data were self-reported, so there may be recall bias or inaccuracies as well as an underestimation of the prevalence of overweight and obesity. In addition, the tendency to respond according to expected social norms may bias self-reports of food consumption. However, the use of self-reported questionnaires is a common method in this type of study, because of their low cost and easy administration; moreover, an instructor was present and available to help [63]. Second, the S-HEI coding system does not provide quantitative information on food and nutrients or measure dietary energy intake; moreover, it is was designed to assess diet quality using 24 h dietary recall scores. However, previous studies in adolescents already assessed diet quality using the FFQ, and the S-HEI scores calculated from it were reliable [64]. Third, a technical error in the survey configuration resulted in 57% of the data on mobile phone use being missed, with the relative question only asked in some schools. Despite this, the variable was considered in the analyses because of its potential relevance, as it is expected that the overall behavior of the sample remains consistent. Fourth, we have used p-values to conclude whether there were statistically significant associations between variables, as they are widely accepted in the scientific community. However, the use of the concept of statistical significance has been under debate in recent years. Finally, the design of this study did not allow us to establish causality for the significant associations studied. 

One of the main strengths of our study consists in providing important information on the quality of adolescents’ diet thanks to a relatively large sample (7319 students from 71% of the secondary education centers of Central Catalonia) which includes public and chartered/private centers, from larger cities and smaller towns, and from different socio-economic levels of the municipality, among other socio-demographic characteristics. If we estimate the power over 15,000 potential participants (number of people attending educational centers in Central Catalonia in the academic year 2019–2020) with a precision unit ± 1 percentile unit and with a confidence level of 95%, we would need a sample of 2811 participants. In our case, the final sample was 2.6 times the required size (7319). This allowed us to perform a strong statistical analysis and find specific associations between variables, as well as to differentiate our study from others by providing information on rural and not only urban populations [12,65,66]. In addition, our data are specific for adolescents, and not mixed with others from the infant/juvenile population, as was the case in other studies. Finally, this study analyzes data from the first wave of a cohort and could be the foundation for future interventions and for a follow-up in the studied population. 

## 5. Conclusions

The diet quality of most of the adolescents from Central Catalonia needs to be improved. The associations between S-HEI and gender, and socioeconomic and health-related factors, highlight the need to plan health promotion programs as an integral part of the education program in schools to avoid increasing nutritional and health inequalities.

## Figures and Tables

**Figure 1 nutrients-16-00139-f001:**
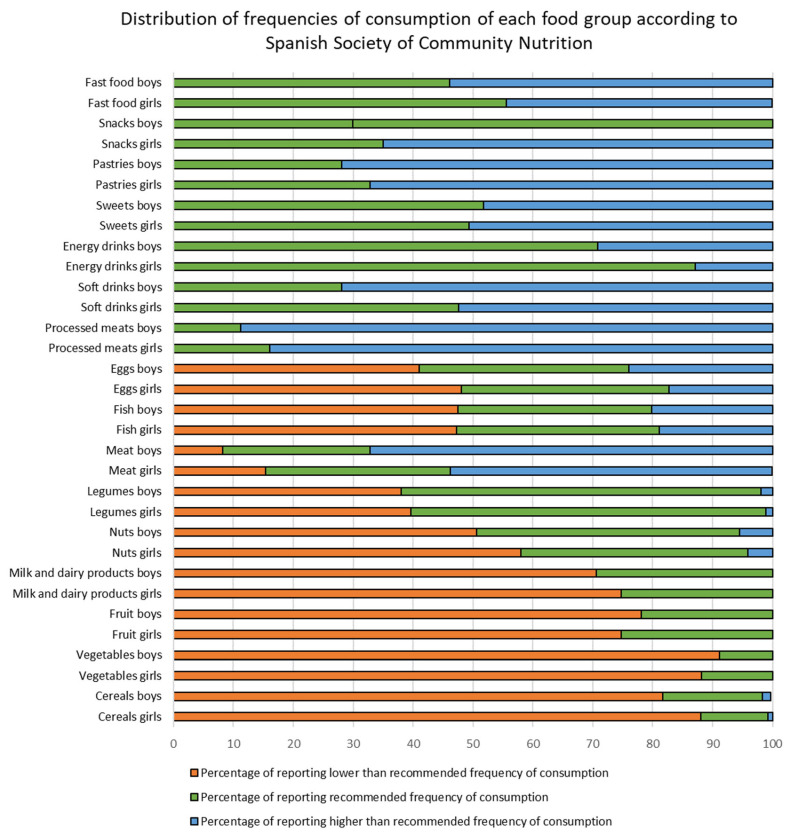
Distribution of frequencies of consumption of each food group according to the Spanish Society of Community Nutrition dietary guidelines by boys and girls.

**Table 1 nutrients-16-00139-t001:** DESK-cohort participant characteristics by gender.

	Total (n = 7319)		Boys (n = 3505)		Girls (n = 3814)		*p*
	n	%	n	%	n	%	
**Age (mean, SD) ***	15.3	0.0	15.3	0.0	15.3	0.0	0.53
**Course**							**<0.001**
2nd course of CSE	2678	36.6	1296	37.0	1382	36.2	
4th course of CSE	2690	36.8	1316	37.5	1374	36.0	
2nd course of PCSE	1512	20.7	649	18.5	863	22.6	
ILTC	439	6.0	244	7.0	195	5.1	
**Size of municipality**							**0.015**
≤5000	2010	27.5	1015	29.0	995	26.1	
5001–20,000	2915	39.8	1350	38.5	1565	41.0	
>20,000	2206	30.1	1059	30.2	1147	30.1	
No data	188	2.6	81	2.3	107	2.8	
**Perceived socioeconomic position ^a^**							0.465
Disadvantaged	2596	35.5	1221	34.8	1375	36.1	
Medium	2419	33.1	1180	33.7	1239	32.5	
Advantaged	2304	31.5	1104	31.5	1200	31.5	
**Parents’ highest level of education**							**<0.001**
University education	2714	37.1	1346	38.4	1368	35.9	
Secondary education	2070	28.3	913	26.1	1157	30.3	
Primary education	1231	16.8	506	14.4	725	19.0	
No data	1304	17.8	740	21.1	564	14.8	
**Body Mass Index (BMI) ^b^**							<0.001
Underweight	212	2.9	120	3.4	92	2.4	
Healthy weight	5496	75.1	2503	71.4	2993	78.5	
Overweight	1042	14.2	590	16.8	452	11.9	
Obesity	287	3.9	170	4.9	117	3.1	
No data	282	3.9	122	3.5	160	4.2	
**Physical activity ^c^**							**<0.001**
Compliance with WHO recommendations	3271	44.7	1968	56.2	1303	34.2	
Under WHO recommendations	3510	48.0	1310	37.4	2200	57.7	
No data	538	7.4	227	6.5	311	8.2	
**Alcohol use ^d^**							**<0.001**
Non-hazardous drinking	6351	86.8	3137	89.5	3214	84.3	
Hazardous drinking	968	13.2	368	10.5	600	15.7	
**Tobacco use**							**0.008**
Others	6768	92.5	3271	93.3	3497	91.7	
Daily use	551	7.5	234	6.7	317	8.3	
**Cannabis use ^e^**							0.13
No risk Consumption	7032	96.1	3355	95.7	3677	96.4	
Risk Consumption	287	3.9	150	4.3	137	3.6	
**Mobile use ^f^**							**<0.001**
No problematic use	1374	18.8	703	20.1	671	17.6	
Occasional or frequent problems	1767	24.1	757	21.6	1010	26.5	
No data	4178	57.1	2045	58.4	2133	55.9	
**Self-percieved health**							**<0.001**
Excellent/very good	4201	57.4	2290	65.3	1911	50.1	
Good	2417	33.0	966	27.6	1451	38.0	
Very poor/poor	701	9.6	249	7.1	452	11.9	
**Mood state ^g^**							**<0.001**
Good mood	5904	80.7	3075	87.7	2829	74.2	
Low mood	1415	19.3	430	12.3	985	25.8	
**Sleep quality**							**<0.001**
Very good/good	5199	71.0	2706	77.2	2493	65.4	
Very poor/poor	2120	29.0	799	22.8	1321	34.6	
**Self-reported academic performance**							**<0.001**
Good grades	2012	27.5	884	25.2	1128	29.6	
Average grades	4244	58.0	2036	58.1	2208	57.9	
Poor grades	720	9.8	413	11.8	307	8.1	
No data	343	4.7	172	4.9	171	4.5	
**Having experienced bullying**							**0.032**
Yes	564	7.7	296	8.5	268	7.0	
No	6413	87.6	3064	87.4	3349	87.8	
No data	342	4.7	145	4.1	197	5.2	
**Diet quality ^h^**							**<0.001**
Healthy diet	602	8.2	186	5.3	416	10.9	
Diet needs changes	6238	85.2	3070	87.6	3168	83.1	
Unhealthy diet	479	6.5	249	7.1	230	6.0	
**Diet Quality score ^i^ (mean, SD) ***	66.4	0.1	65.1	0.2	67.6	0.2	**<0.001**

Statistically significant associations are highlighted in bold in the p-value columns. Abbreviations: SD = Standard Deviation; 2nd and 4th course of CSE (Compulsory Secondary Education) corresponds to ISCED 2 (International Standard Classification of Education); 2nd PCSE (Post Compulsory Secondary Education) and ILTC (Intermediate Level Training Cycles) corresponds to ISCED3 (International Standard Classification of Education); WHO = World Health Organization; “*p*” Pearson chi2 and Student’s *t* test, *p* < 0.05 were considered statistically significant. * In “Age” and in “Diet Quality score” variable, the heading “n” is substituted with “mean” and “%” with “SD”. ^a^ MacArthur Scale of Subjective Social Status and dividing the participants in tertiles. ^b^ Age- and sex-specific BMI cut-off points according to the WHO growth reference for school-aged children and adolescents. ^c^ In compliance with the WHO recommendation of ≥60 min per day, or under the WHO recommendation of 60 min per day, estimated from the average daily minutes of moderate or vigorous physical activity reported by the adolescents. ^d^ Scores above 3 on the Alcohol Use Disorders Identification Test (AUDIT-C test) were considered hazardous drinking. ^e^ Scores above 7 on the Cannabis Abuse Screening Test (CAST test) were considered risk consumption. ^f^ Scores above 15 on the Questionnaire for mobile phone-related experiences (CERM test) were considered problematic mobile use. ^g^ Scores of 3 or more was identified as a low mood. ^h^ Unhealthy diet (<50 points); diet that needs changes (50–80 points); and healthy diet (>80 points). ^i^ Healthy Eating Index Spanish adaptation (S-HEI index score).

**Table 2 nutrients-16-00139-t002:** Mean S-HEI scores by socioeconomic factors and health behaviors within gender (n = 7319).

	Boys S-HEI (n = 3505)			Girls S-HEI (n = 3814)		
	M	95% CI	*p*	M	95% CI	*p*
**Age (r)**	−0.3	(−0.5; 0.0)	**0.02**	0.0	(−0.2; 0.3)	0.68
**Course**			**<0.001**			**<0.001**
2nd course of CSE	65.8	(65.2; 66.3)		67.5	(67.0; 68.1)	
4th course of CSE	64.5	(63.9; 65.0)		67.2	(66.6; 67.8)	
2nd course of PCSE	66.5	(65.8; 67.3)		69.7	(69.0; 70.3)	
ILTC	61.4	(60.1; 62.7)		62.6	(60.9; 64.3)	
**Size of municipality**			0.05			0.06
≤5000	65.1	(64.5; 65.8)		68.1	(67.4; 68.7)	
5001–20,000	64.8	(64.2; 65.3)		67.3	(66.8; 67.8)	
>20,000	65.8	(65.2; 66.4)		68.2	(67.6; 68.8)	
**Perceived socioeconomic position ^a^**			**0.02**			**<0.001**
Disadvantaged	64.3	(63.7; 64.9)		66.5	(65.9; 67.1)	
Medium	65.7	(65.1; 66.2)		67.8	(67.3; 68.4)	
Advantaged	65.5	(64.9; 66.1)		68.7	(68.1; 69.3)	
**Parents’ highest level of education**			**<0.001**			**<0.001**
University education	67.2	(66.7; 67.7)		70.3	(69.8; 70.8)	
Secondary education	64.6	(64.0; 65.3)		67.5	(66.9; 68.1)	
Primary education	63.7	(62.8; 64.6)		64.8	(63.9; 65.6)	
No data	63.0	(62.1; 63.8)		65.2	(64.3; 66.1)	
**Body Mass Index ^b^**			0.54			0.67
Underweight	64.2	(62.1; 66.2)		67.2	(65.0; 69.5)	
Healthy weight	65.2	(64.8; 65.6)		67.8	(67.4; 68.5)	
Overweight	65.6	(64.8; 66.4)		67.6	(66.6; 68.5)	
Obesity	64.7	(63.1; 66.2)		66.5	(64.8; 68.3)	
No data	63.6	(61.7; 65.6)		66.1	(64.2; 67.9)	
**Physical activity ^c^**			**<0.001**			**<0.001**
Compliance with WHO recommendations	66.0	(65.5; 66.4)		69.5	(68.9; 70.0)	
Under WHO recommendations	64.4	(63.8; 64.9)		67.1	(66.6; 67.6)	
No data	62.4	(60.8; 64.0)		63.9	(62.6; 64.1)	
**Alcohol use ^d^**			**<0.001**			**<0.001**
Non-hazardous drinking	65.5	(65.1; 65.8)		68.0	(67.6; 68.4)	
Hazardous drinking	62.1	(61.1; 63.2)		65.8	(64.8; 66.7)	
**Tobacco use**			**<0.001**			**<0.001**
Others	65.5	(65.2; 65.8)		68.1	(67.7; 68.5)	
Daily use	60.0	(58.5; 61.5)		62.7	(61.4; 64.0)	
**Cannabis use ^e^**			**<0.001**			**<0.001**
No risk Consumption	65.3	(65.0; 65.7)		67.9	(67.6; 68.2)	
Risk Consumption	60.6	(58.7; 62.5)		60.8	(58.7; 63.0)	
**Mobile use ^f^**			**<0.001**			**<0.001**
No problematic use	66.7	(66.0; 67.5)		69.1	(68.4; 69.9)	
Occasional or frequent problems	64.0	(63.3; 64.7)		66.1	(65.4; 66.8)	
No data	65.0	(64.6; 65.4)		67.9	(67.5; 68.4)	
**Self-perceived health**			**<0.001**			**<0.001**
Excellent/very good	66.1	(65.7; 66.5)		69.7	(69.2; 70.1)	
Good	63.5	(62.9; 64.2)		66.4	(65.8; 67.0)	
Very poor/poor	62.7	(61.2; 64.1)		63.2	(62.0; 64.3)	
**Mood state ^g^**			**0.02**			**<0.001**
Good mood	65.3	(64.9; 65.6)		68.3	(67.9; 68.7)	
Low mood	64.1	(63.1; 65.1)		65.8	(65.0; 66.5)	
**Sleep quality**			**0.001**			**0.008**
Very good/good	65.4	(65.0; 65.8)		68.0	(67.6; 68.4)	
Very poor/poor	64.1	(63.4; 64.9)		66.7	(66.4; 67.6)	
**Academic performance**			**<0.001**			**<0.001**
Good grades	67.8	(67.2; 68.4)		70.6	(70.1; 71.2)	
Average grades	64.8	(64.4; 65.3)		67.1	(66.7; 67.6)	
Poor grades	62.1	(61.1; 63.1)		62.2	(60.8; 63.5)	
No data	62.5	(60.8; 64.2)		64.6	(62.8; 66.5)	
**Having experienced bullying**			**0.004**			**<0.001**
No	65.4	(65.0; 65.7)		67.9	(67.5; 68.3)	
Yes	63.7	(62.4; 65.1)		65.3	(63.8; 66.7)	
No data	63.5	(61.7; 65.2)		66.5	(65.0; 68.1)	

Statistically significant associations are highlighted in bold in the p-value columns. Abbreviations: 2nd and 4th course of CSE (Compulsory Secondary Education) corresponds to ISCED 2 (International Standard Classification of Education); 2nd PCSE (Post Compulsory Secondary Education) and ILTC (Intermediate Level Training Cycles) corresponds to ISCED3 (International Standard Classification of Education); WHO = World Health Organization; S-HEI = Healthy Eating Index Spanish adaptation; M = Mean; CI = Confidence Interval; “*p*” Analysis of variance ANOVA—F test, *p* < 0.05 were considered statistically significant. ^a^ MacArthur Scale of Subjective Social Status and dividing the participants in tertiles. ^b^ Age- and sex-specific BMI cut-off points according to the WHO growth reference for school-aged children and adolescents. ^c^ In compliance with the WHO recommendation of ≥60 min per day, or under the WHO recommendation of 60 min per day, estimated from the average daily minutes of moderate or vigorous physical activity reported by the adolescents. ^d^ Scores above 3 on the Alcohol Use Disorders Identification Test (AUDIT-C test) were considered hazardous drinking. ^e^ Scores above 7 on the Cannabis Abuse Screening Test (CAST test) were considered risk consumption. ^f^ Scores above 15 on the Questionnaire for mobile phone-related experiences (CERM test) were considered problematic mobile use. ^g^ Scores of 3 or more was identified as a low mood.

**Table 3 nutrients-16-00139-t003:** Multivariable regression results for S-HEI score by gender (n = 7319).

	Boys S-HEI (n = 3505)			Girls S-HEI (n = 3814)		
	Adjusted Coef.	95% CI	*p*	Adjusted Coef.	95% CI	*p*
**Course**						
2nd course of CSE	0			0		
4th course of CSE	−1.1	(−1.8; −0.3)	**0.006**	0.3	(−0.5; 1.1)	0.43
2nd course of PCSE	1.2	(0.2; 2.2)	**0.02**	2.9	(1.9; 3.8)	**<0.001**
ILTC	−2.9	(−4.3; −1.5)	**<0.001**	−1.9	(−3.5; −0.3)	**0.02**
**Parents’ highest level of education**						
University education	0			0		
Secondary education	−1.7	(−2.5; −0.9)	**<0.001**	−1.8	(−2.7; −1.0)	**<0.001**
Primary education	−2.1	(−3.1; −1.1)	**<0.001**	−3.3	(−4.3; −2.3)	**<0.001**
No data	−3.3	(−4.2; −2.4)	**<0.001**	−3.2	(−4.2; −2.2)	**<0.001**
**Physical activity ^a^**						
Compliance with WHO recommendations	0			0		
Under WHO recommendations	−0.9	(−1.6; −0.2)	**0.01**	−1.2	(−1.9; −0.4)	**0.002**
No data	−2.5	(−3.8; −1.1)	**<0.001**	−3.5	(−4.8; −2.2)	**<0.001**
**Alcohol use ^b^**						
Non-hazardous drinking	0			0		
Hazardous drinking	−2.5	(−3.7; −1.3)	**<0.001**	−1.0	(−1.9; 0.0)	**0.05**
**Tobacco use**						
Others	0			0		
Daily use	−3.1	(−4.5; −1.6)	**<0.001**	−2.6	(−3.9; −1.3)	**<0.001**
**Mobile use ^c^**						
No problematic use	0			0		
Occasional or frequent problems	−2.3	(−3.4; −1.3)	**<0.001**	−2.0	(−3.1; −1.0)	**<0.001**
No data	−1.5	(−2.3; −0.7)	**<0.001**	−0.6	(−1.5; 0.3)	0.18
**Self-perceived health**						
Excellent/very good	0			0		
Good	−1.1	(−1.9; −0.4)	**0.003**	−1.8	(−2.5; −1.1)	**<0.001**
Very poor/poor	−1.1	(−2.4; 0.2)	0.09	−3.5	(−4.6; −2.4)	**<0.001**
**Mood state ^d^**						
Good mood				0		
Low mood				−1.1	(−1.9; −0.3)	**0.006**
**Academic performance**						
Good grades	0			0		
Average grades	2.1	(−2.9; −1.3)	**<0.001**	−2.1	(−2.9; −1.4)	**<0.001**
Poor grades	−3.9	(−5.1; −2.8)	**<0.001**	−5.2	(−6.6; −3.9)	**<0.001**
No data	−3.5	(−5.1; −1.9)	**<0.001**	−3.4	(−5.1; −1.7)	**<0.001**

Statistically significant associations are highlighted in bold in the p-value columns. Abbreviations: 2nd and 4th course of CSE (Compulsory Secondary Education) corresponds to ISCED 2 (International Standard Classification of Education); 2nd PCSE (Post Compulsory Secondary Education) and ILTC (Intermediate Level Training Cycles) corresponds to ISCED3 (International Standard Classification of Education); WHO = World Health Organization; S-HEI = Healthy Eating Index Spanish adaptation; CI = Confidence Interval; “*p*” Student’s *t* test, *p* < 0.05 were considered statistically significant. ^a^ In compliance with the WHO recommendation of ≥60 min per day, or under the WHO recommendation of 60 min per day, estimated from the average daily minutes of moderate or vigorous physical activity reported by the adolescents. ^b^ Scores above 3 on the Alcohol Use Disorders Identification Test (AUDIT-C test) were considered hazardous drinking. ^c^ Scores above 15 on the Questionnaire for mobile phone-related experiences (CERM test) were considered problematic mobile use. ^d^ Scores of 3 or more was identified as a low mood.

## Data Availability

The data presented in this study are available upon reasonable re-quest to the corresponding author. The data are not publicly available due to confidentiality reasons.

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
