# Peer review of "Relationship between Diet Quality and Socioeconomic and Health-Related Factors in Adolescents by Gender"

_nutrients, 2023, doi:10.3390/nu16010139_

Round 1

Reviewer 1 Report

Comments and Suggestions for Authors

The paper addresses an important issue and is based on valuable information. However, there are several limitations that need to be addressed:

Lines 53 – 55: The authors should address the issue of comparing HEI data “between regions and countries”, given the fact that the HEI has been modified over time and was adapted to regional or national situations.

Lines 77 – 78: The definition of urban and rural areas is not clear.

Lines 94 – 97: The authors should provide more information regarding the applied FFQ: Which time period does it cover (one week, one month, or one year)? Is it a quantitative or a semi-quantitative questionnaire? Was the questionnaire validated?

Lines 100 – 104: The authors should provide more information regarding the S-HEI, since the given reference is not available in the English language. In which way does S-HEI differ from the original HEI suggested by Kennedy et al.? Why did the authors not refer to a more recent version of the HEI. What is the range of S-HEI values? It seems they range from 0 to 110. Please confirm or clarify.

Lines 126 – 128: It seems that the S-HEI is purely based on eating frequencies without any quantitative considerations. This would be a substantial deviation from the original HEI and pose a serious limitation of the provided data. Please clarify!

Lines 131 – 136: The definition of the level of education is confusing. Please provide a brief explanation.

Lines 136 – 162: The authors apply several assessment tools such as SEP, AUDIT-C, CAST, etc. Are those tools truly suitable for adolescents?

Lines 141 – 145: The categorization of BMI values is confusing. Maybe, the authors should base their definitions on z-scores. I suppose there is no difference in the meaning of “sd” and “SD”? The reference to BMI categorization in adults (“at 19 years”) is unnecessarily confusing and should be omitted.  

Lines 148 – 153: Did the authors use any tools to assess “self-perceived health” and “mood state”?

Lines 149 – 153: What is the reasoning for considering the state of the mood at a certain time point in relation to habitual food consumption as assessed by an FFQ?

Lines 153 – 158: The authors should discuss the expected validity of provided information regarding the consumption of alcohol, tobacco, and cannabis, particularly in the context of a school setting. I would expect a severe bias.

Lines 69 – 162: The authors should provide information on how many questions did the subjects have to answer in total. What was the time span required for providing all the information?

Lines 163 – 177: Two sample t-tests are only appropriate if variables are normally distributed. How did the authors make sure?

Lines 163 – 177: What is the theoretical background of the regression model? How can the model be justified? The paper gives the impression that the authors used a mere conglomeration of independent variables rather than a true model. This is unpinned by the fact that the authors did not consider interactions between their independent variables.

Lines 171 – 173: Why did the authors use backward elimination?

Lines 163 – 177: The authors should acknowledge that “significance” is an outdated concept in statistics (c.f. doi: https://doi.org/10.1038/d41586-019-00857-9 or https://doi.org/10.1080/00031305.2019.1583913 ). Therefore, they should adjust their statistical analyses and their interpretations accordingly.

Lines 163 – 177: Defining “no data” as a separate category in inferential statistics may lead to confusing results. The obtained p-values and the interpretation of the results might be heavily influenced by those missing data. Therefore, the category “no data” should be eliminated, particularly for the analyses leading to the results in Table 2.

Lines 178 – 181: What was the dropout rate? What was the percentage of subjects who provided all the requested information?

Line 183: A differentiation between pre-obese and obese subjects would grant valuable insight.

Line 271 – 400: The discussion is quite lengthy. On several occasions, the authors point out that their results confirm previous studies. But to what extent does the study provide new insight into this important issue?

Line 271 – 400: On several occasions, the authors draw conclusions that go way beyond the scope of their data (e.g., lines 374-79).

Lines 380 – 391: The study bears a lot more limitations than those pointed out by the authors. Some of them are given in this review and should be included in the discussion.

Figure 1: Merging the two graphs would ease the comparison of girls and boys. Furthermore, the legend in Figure 1.b is incomplete.

Reviewer 2 Report

Comments and Suggestions for Authors

The article takes up an interesting topic. Below I post some comments and suggestions to improve the manuscript as well as the questions for the authors.

Abstract

It should be supplemented with: date of the study and description of the study group (size and  age of the subjects), there is no information about the research tool and the data collection method.   

Introduction

In general - this part is well written and in my opinion does not need improvement, only one suggestion - line 38 – need reference/s.

Material and methods

In general - this part is well written and in my opinion does not need improvement.

Was the study group representative to some extent? If so, for which community?

Results

In general - in the results section is well written but I have a few suggestion, as below:

Figure 1 is well prepared and clear, but if someone (another author) wants to use and quote this data, e.g. in his own discussion, it lacks specific values. I recommend that the authors prepare this data (e.g. in the form of a table) to be included in as a supplementary material.

Discussions   

This section is well written but in the limitations of the study and strength sections there is no information about the representativeness of the group - the authors only wrote that it was a large study group and it is not the same.

Conclusions

The conclusions in the abstract and at the end of the paper should be compatible as well as cannot be too general because it is not known whether the study group is representative? The answer to this question is important - if it is not a representation (?), the results cannot be generalized and must refer only to the studied group.

Reviewer 3 Report

Comments and Suggestions for Authors

Dear Authors, the following remarks and comments are intended to increase the readability of the manuscript for a wide range of readers:

The abstract is not informative enough: basic information is missing, i.e. the number of study participants and their age. The authors refer in this part to another study (DESKcohort project), but the reader does not need to be familiar with this data.

Introduction

The first sentence lacks a reference to the literature. Not all kind of cancers are diet/nutrients related and not all type of diabetes (type I is not diet-related). Authors should be more precise. In this part, I lack information about the nutrition/nutritional habits of adolescents in Spain, especially in terms of problems with balancing the diet. Do adolescents in Spain have any specific eating behaviors or their diet is characterized by typical mistakes for this age group (snacks, sweetened drinks, sweets, irregularities, etc.).

Methods

Lines 116-118: On what basis was the S-HEI "healthiness" point classification adopted?

The abbreviation “ISCED” is not explained i n the text.

Were the tests (AUDIT-C, CAST ect.) used in the study validated on Spanish adolescents? This information is missing.

Table 1: I cannot see SD for age?

Figure 1: No description for the blue bar for girls – please check.

Discussion

This part is correctly laid out and legible. The authors drew attention to the limitations of the study related to the method of data collection. Have the Authors tried to estimate the error in obtaining data on adolescents' weight and height by CAWI method (without measurements)? Are the obtained results regarding anthropometric parameters significantly different compared to data representative of adolescents in Spain obtained from anthropometric measurements? Based on their own experience or observations, can they assume that the data provided by respondents was, for example, over- or underestimated?

In this part, the authors could consider additional literature on the topic: https://doi.org/10.3390/nu12051323.
